# Physico-Chemical Investigation of Endodontic Sealers Exposed to Simulated Intracanal Heat Application: Hydraulic Calcium Silicate-Based Sealers

**DOI:** 10.3390/ma14040728

**Published:** 2021-02-04

**Authors:** David Donnermeyer, Magdalena Ibing, Sebastian Bürklein, Iris Weber, Maximilian P. Reitze, Edgar Schäfer

**Affiliations:** 1Department of Periodontology and Operative Dentistry, Westphalian Wilhelms-University, Albert-Schweitzer-Campus 1, Building W 30, 48149 Münster, Germany; magdalena.ibing@ukmuenster.de; 2Central Interdisciplinary Ambulance in the School of Dentistry, Albert-Schweitzer-Campus 1, Building W 30, 48149 Münster, Germany; sebastian.buerklein@ukmuenster.de (S.B.); eschaef@uni-muenster.de (E.S.); 3Institute for Planetology, Westphalian Wilhelms-University, Wilhelm-Klemm-Str. 10, 48149 Münster, Germany; sonderm@uni-muenster.de (I.W.); maximilian.p-reitze@wwu.de (M.P.R.)

**Keywords:** BioRoot RCS, calcium silicate, calcium silicate-based sealer, intracanal heat application, warm vertical obturation, Total Fill BC Sealer, Total Fill BC Sealer HiFlow, Fourier-transform infrared spectroscopy, ISO 6876, root canal filling materials, root canal obturation

## Abstract

The aim of this study was to gain information about the effect of thermal treatment of calcium silicate-based sealers. BioRoot RCS (BR), Total Fill BC Sealer (TFBC), and Total Fill BC Sealer HiFlow (TFHF) were exposed to thermal treatment at 37 °C, 47 °C, 57 °C, 67 °C, 77 °C, 87 °C and 97 °C for 30 s. Heat treatment at 97 °C was performed for 60 and 180 s to simulate inappropriate application of warm obturation techniques. Thereafter, specimens were cooled to 37 °C and physical properties (setting time/flow/film thickness according to ISO 6876) were evaluated. Chemical properties (Fourier-transform infrared spectroscopy) were assessed after incubation of the specimens in an incubator at 37 °C and 100% humidity for 8 weeks. Statistical analysis of physical properties was performed using the Kruskal-Wallis-Test (*P* = 0.05). The setting time, flow, and film thickness of TFBC and TFHF were not relevantly influenced by thermal treatment. Setting time of BR decreased slightly when temperature of heat application increased from 37 °C to 77 °C (*P* < 0.05). Further heat treatment of BR above 77 °C led to an immediate setting. FT-IR spectroscopy did not reveal any chemical changes for either sealers. Thermal treatment did not lead to any substantial chemical changes at all temperature levels, while physical properties of BR were compromised by heating. TFBC and TFHF can be considered suitable for warm obturation techniques.

## 1. Introduction

Calcium silicate-based sealers were introduced to endodontics more than a decade ago [1]. In addition, other calcium silicate-based materials for direct pulp capping [2] and root end filling [3] have become standard in modern dentistry. Whilst calcium silicate-based sealers display a new material class incorporating beneficial properties from calcium silicate cements, these new sealers attempted to change a dogma of root canal obturation. Previously, a relatively high proportion of a core material such as gutta-percha and a small amount of a root canal sealer were the desired outcome of root canal filling. Calcium silicate-based sealers though were designed for cold obturation techniques, especially for single cone root canal filling [4]. Higher sealer proportions were no longer considered disadvantageous, as biological and antibacterial effects of calcium-silicate based sealers are thought to enhance the success of root canal treatment. Due to a low compression of the root canal filling material and sealer during single cone root canal filling, this technique is regarded as incapable of filling complex root canal systems [5]. In simpler root canal anatomies, it might allow sufficient root canal filling [6]. Warm root canal obturation techniques were invented to allow three-dimensional filling of complex root canal systems [5,7]. To fill canal irregularities and potentially lateral canals, gutta-percha is thermoplasticized inside or outside the root canal and inserted into the canal. This leads to a lower proportion of root canal sealer within the root canal filling. Still, the sealer is necessary to seal the root canal system at the root canal wall [8].

To allow the use of warm obturation techniques together with calcium silicate-based sealers, a new sealer—Total Fill BC Sealer HiFlow (TFHF; FKG Dentaire, La Chaux des Fonds, Switzerland)—was recently introduced [9]. It is labelled to fulfil all requirements for a sealer to be used with warm vertical root canal filling techniques. Previously, no calcium silicate-based sealer suitable for warm vertical compaction techniques was commercially available and warm obturation techniques were limited to other sealer types such as epoxy resin- or zinc oxide eugenol-based. These sealers were already proven to be stable when exposed to intracanal heat application [10]. Other calcium silicate-based sealers like Total Fill BC Sealer (TFBC; FKG Dentaire) or BioRoot RCS (BR; Septodont, St. Maur-des-Fossés, France) have recently been investigated regarding their behavior when exposed to heat application. Physical and chemical properties were investigated, during or shortly after the application of heat [11,12,13,14,15]. While the physical properties of TFHF were not negatively affected by heat, BR and TFBC were reported for increased viscosity [11]. Chemical changes were found in TFBC during the thermal treatment, but reported to remiss after the cooling process while BR was reported as having persisting chemical changes [12]. Shortly after setting, no alterations were reported for TFHF, TFBC, and BioRoot RCS after initial thermal treatment [11]. Though the setting reaction of calcium silicate-based materials is a process of several weeks [16], no data are, so far, available that address the long-term effect of thermal treatment on calcium silicate-based sealers. In addtion, the heat treatments in some previous studies were rather simple, as specimens were treated in ovens for a certain period at only a few temperature levels. Lately, a more decisive and clinically based thermal treatment was presented in closed containers with thermographic control inside the sealer specimen to follow the clinical situation at its best [10,17].

The aim of the this laboratory study was to assess, under simulated clinical conditions, the effect of clinically orientated thermal treatment on selected physical properties immediately after heating and on the long-term chemical properties of calcium silicate-based sealers. The null hypothesis tested was that duration of heat application and range of temperature have no relevant impact on the physical (setting time, flow, film thickness) or chemical properties of the calcium silicate-based sealers BioRoot RCS, Total Fill BC Sealer, and Total Fill BC Sealer HiFlow.

## 2. Materials and Methods

Table 1 reveals the composition of the sealers as declared by the manufacturers. To allow comparison of the results, the method was applied in a similar manner to a previously published study on thermal effects on sealer [10].

### 2.1. Heat Application on the Sealers

On a glass plate, BR was mixed in strict accordance to the instructions provided by the manufacturer. TFBC and TFHF were dispensed directly from the syringe. Portions of 0.5 mL of each freshly mixed sealer were encapsulated into a 2 mL plastic tube (Safe-Lock Tubes, Eppendorf, Hamburg, Germany). Inside the samples, a K-type thermocouple (GHM Messtechnik, Regenstauf, Germany) was placed, and the samples were heated in a thermo-controlled water bath until the temperatures of 37 °C, 47 °C, 57 °C, 67 °C, 77 °C, 87 °C and 97 °C were achieved inside the samples (Figure 1). These temperatures were selected in accordance with recently published data [17], as the maximum intracanal temperatures reported were 19.1 °C above body temperature. Furthermore, a wide range of temperatures was selected to assess the impact of overheating the sealers. The temperature of the sealer was checked using the mentioned K-Type thermocouple and the GSVmulti software (version 1.27, ME-Meßsysteme, Hennigsdorf, Germany) at a frequency of 50 Hz during the whole procedure to guarantee controlled warming of the sealers. Thereafter, all samples were retained for 30 s at the respective temperature and were then cooled to 37 °C in a second water bath. In order to evaluate the influence of elongated heating, which may result when warm vertical compaction techniques are inappropriately implemented, samples were also heated to 97 °C for 60 and 180 s [18]. After cooling the samples, they were exposed to testing of the physical properties or kept on glass plates in an incubator for 8 weeks at 37 °C and 100% humidity until further evaluation of their chemical characteristics (Figure 1). The described procedure took about 3 min until the samples were analysed further.

### 2.2. Evaluation of Physical Characteristics

#### 2.2.1. Setting Time

According to ISO specification 6876:2012 (International Standard ISO 6876:2012), the sealers’ setting time was evaluated [19]. The preheated sealer specimens were dispensed into plaster molds (d = 10 mm, h = 1 mm) and transferred to an incubator at 37 °C and 100 % humidity. Using a stopwatch, the setting time of the material was measured. A cylindrical indenter with a flat end tip diameter of 2 mm and a mass of 100 g was used in accordance to ISO 6876. During the first hour after mixing, this procedure was repeated at 2-min intervals and thereafter at 30-min intervals. The setting point of the materials was defined as the point when the needle left no indentation on its surface. For each temperature level and each sealer, three iterations were performed and the mean was computed.

#### 2.2.2. Film Thickness

The film thickness of the preheated samples was assessed in accordance to ISO specification 6876:2012, but due to the temporal process when preheating the sealers, minor adjustments were necessary. A small amount of each preheated sealer was situated on a glass plate (dimensions: 40 mm × 40 mm; thickness 5 mm). Another glass plate with the same thickness, but a smaller surface area (200 mm^2^), was placed on top centrally. This second glass plate was loaded vertically for 10 min with 150 N using an universal testing machine (Lloyd LF Plus, Ametek, Berwin, PA, USA). Before each evaluation, the total thickness of the two assembled glass plates was determined using a digital micrometer and this was repeated after the testing procedure. For each group (temperature level and type of sealer), the measurements were performed in triplicate and the mean was computed.

#### 2.2.3. Flow

Following thermal treatment, the sealers’ flow was determined in accordance to ISO specification 6876:2012 with minor modifications. Because of the increased viscosity of the sealers at high temperatures, which was found in a preliminary study [10], the sealers were portioned by weight instead of volume. With the use of a precision scale and a graduated pipette, it was determined that 0.05 mL of sealer corresponded to 0.1265 g of Total Fill BC Sealer, 0.1276 g of Total Fill BC Sealer HiFlow, and 0.110 g of BioRoot RCS, respectively at 20 °C. All sealer portions were measured out by weight during the investigation. A certain amount of each specimen was placed on a glass plate with the same dimensions as described above. An identical second glass plate was placed on top centrally and the total mass was 120 g. This assembly was kept for 10 min. Using a digital caliper, the maximum and minimum diameters of the compressed sealer disc were measured. In the case the maximum and minimum diameters were within 1 mm, the mean was computed. In the case of greater differences, the measurement was repeated. For each group (temperature level and type of sealer), the measurements were performed in triplicate and the mean was computed.

### 2.3. Evaluation of Chemical Characteristics

Fourier Transform Infrared Spectroscopy (FT-IR)

For FT-IR analysis of the preheated sealers, the set specimens were powdered using a mortar. A 0.002 g increment of sealer powder of each specimen was added to 0.200 g potassium bromide and pressed to a pill (13 mm in diameter) with the help of a pellet die (Specac, Orpington, UK) and a manual laboratory compactor (Type PW 10, Paul-Otto Weber Laborpresstechnik, Remshalden, Germany). FT-IR was conducted using the Vertex 70v (Bruker, Billerica, MA, USA) with a mercury cadmium telluride (MCT) detector and the related software OPUS (Bruker). Background spectra of KBR were collected. Two tests (256 scans per test) with two different samples were performed for each temperature level to control the FT-IR analysis. As long as no difference between these two samples were evident, one result was selected for further graphic illustration and assessment.

### 2.4. Statistical Analysis

Data of the physical properties (setting time, film thickness and flow) were analyzed using the Kruskal-Wallis-Test at *P* = 0.05, applying MedCalc (Version 19.5.6, MedCalc Software, Ostende, Belgium). Bonferroni adjustment was performed to allow post-hoc multiple comparisons.

## 3. Results

Differences between the sealers regarding the time period required to attain the determined temperatures were not observed.

### 3.1. Impact of Heat Application on Physical Characteristics

#### 3.1.1. Setting Time

The setting time of BR diminished slightly when temperature of heat application increased from 37 °C to 77 °C (*P* < 0.05). Further heat treatment of BR above 77 °C resulted in an immediate setting of the sealer (Table 2).

The setting time of TFBC varied at about 17 to 19 h, whereas setting time of TFHF was approximately 22 h at all temperatures. Setting time of TFBC (Table 3) and TFHF (Table 4) did not reach a clinically relevant minimum threshold.

#### 3.1.2. Film Thickness

The film thickness of BR varied constantly at approximately 8 µm at temperatures of 37 °C to 77 °C. Further heat treatment of BR above 77 °C caused an immediate setting of the material and it was impossible to measure the film thickness (Table 2). 

Film thickness of TFBC and TFHF varied at approximately 8 µm and was in accordance with the ISO 6876:2012 standard at all temperatures (Table 3 and Table 4).

#### 3.1.3. Flow

The flow rates of TFBC and TFHF were above 17 mm as required by the ISO 6876:2012 specification at all temperatures tested (Table 3 and Table 4). In addition, the flow rates of BR were within these limitations up to a temperature level of 77 °C (Table 2). Further heat treatment of BR caused an immediate setting of the sealer, and it was not possible to assess the flow rate.

Results of the statistical analysis are summarized in Table 2, Table 3 and Table 4. No statistically significant differences occurred for film thickness of BR and TFHF as well as for flow of TFBC. Though statistically significant differences were found for setting time and partly for flow and film thickness, no pattern corresponding with the thermal treatment could be observed.

### 3.2. Impact of Heat Application on Chemical Characteristics

FT-IR spectroscopy revealed that independent of temperatures used and heating times chemical alterations of the sealers did not result, as no differences in the spectroscopic plots resulted after thermal treatment (Figure 2, Figure 3 and Figure 4). TFBC and TFHF revealed similar spectra with differences in the intensity of absorption.

A broad absorption band at 3400 cm^−1^ and a peak at 1650 cm^−1^ was detected in all sealers and at all temperature levels, indicating the presence of water in the specimens [13]. No characteristic Ca(OH)_2_ bands (O-H stretch at 3646 cm^−1^) were detected in TFHF and TFBC, while a small peak occurred in BR samples in this region [20]. Characteristic bands of carbonates were detected at 878 and from 1400 to 1500 cm^−1^ in all specimens [21,22]. Absorption intensities from 970 to 1000 cm^−1^ were observed in all specimens, indicating the formation of calcium silicate hydrate [21]. Two peaks were observed in TFBC and TFHF at ~2874 cm^−1^ and 2923 cm^−1^, which were assigned to the symmetric stretching of -CH_3_ and C-H-stretching of -CH_2_-, respectively [23,24]. No such bands were observed in BR specimens.

## 4. Discussion

The effect of heat on calcium silicate-based sealers was evaluated at different temperatures. The experimental setup regarding temperature levels and heating times was based on recently reported clinically relevant values [7,17], supplemented by a range of temperatures that could occur. BR was found to set when heated above 77 °C, while heating did not compromise the physical properties of TFBC and TFHF. However, the chemical structures of the calcium silicate-based sealers tested were not negatively affected. Therefore, the null hypothesis was partly accepted.

As already presented in a preliminary study [10], applying clinically relevant heating intervals and maximum temperatures are prerequisites when investigating the effect of heat on endodontic sealers under laboratory conditions. Mimicking the in vivo temperature dissipation, intracanal temperatures ranging below 60 °C during warm vertical compaction approaches were found [17]. The use of carrier-based warm obturation techniques resulted in even lower intracanal temperature rises affecting the sealer with temperatures of less than 10 °C [17]. These facts were taken into consideration in the present study in order to generate clinically relevant results about thermal treatment of root canal sealers. Therefore, sealers were heated over short clinically relevant periods in closed containers and cooled down to body temperature immediately afterwards. In addition, temperature levels were chosen according to clinically based considerations [10,17]. To investigate the physical properties of sealers, assessments according to the ISO specification 6876 are well established [25]. Assessment of chemical changes in sealers after thermal treatment by means of FT-IR spectroscopy presents a well-established methodology [10,14]. To investigate the long-term effect of thermal treatment on calcium silicate-based sealers, specimens were stored for 8 weeks. Calcium silicate-based materials have a prolonged setting reaction [16] and, especially in the case of premixed sealer formulations, communicate with the environment to incorporate surrounding water during the setting reaction [26]. Therefore, further to the short-term effect of heat on possible destruction of the chemical composition the long-term effect of heat on the setting profile is of scientific interest.

Originally, calcium silicate-based sealers were designed for cold obturation techniques [1]. As a growing demand was postulated by clinicians to combine the advantageous properties of calcium silicate-based sealers [4] with three-dimensional obturation of the root canal space using thermo-plasticized gutta-percha, Total Fill BC Sealer HiFlow was especially designed for this purpose. While the effect of heat on this sealer and partly on other calcium silicate-based sealers has already been addressed [11,13,14], the so obtained results might not allow a general recommendation for the clinician as to which sealer can be safely used in combination with a certain warm root canal filling technique. Different temperature levels relying on the heat carrier’s device display [11] or general observations of the heat carrier tip temperature [14] were investigated. Recently, the factual heat carrier’s tip temperature was shown to remain under the displayed temperature levels [27] and, due to temperature dissipation, the intracanal sealer temperature is generally lower than the heat carrier’s temperature [17]. To address the outcome of clinical orientated thermal processing, a range of temperature was chosen [10]. In addition, the long-term effect of thermal processing on the setting reaction was addressed, as studies so far only assessed the effect of heat within the moment of thermal treatment [12] or after the primary setting reaction [11,13]. As prolonged hydration and structure maturation of calcium silicate-based materials have been reported [16], the effect of thermal treatment on the final constitution of the sealers has remained unknown so far.

The sealers investigated in this study showed different behavior after thermal processing. While no relevant changes of the physical properties were found with either TFBC or TFHF, BR fully set after exposure to temperatures of 87 °C and above. In compliance with previously published results [11,13,14], thermal treatment did not affect premixed calcium silicate-based sealer composition. Though TFHF exhibited higher flow and higher setting time at all temperature levels, flow, and setting time both of TFHF and TFBC did not descend under clinically relevant thresholds. Film thickness of TFHF and TFBC appeared equal and was not influenced by temperature (Table 3 and Table 4). Other than the premixed formulation, BR was affected in its physical properties by thermal treatment. Beginning with a temperature of 87 °C, the sealer appeared set and assessment of its physical properties was impossible (Table 2). Aksel et al. [11] also reported a lower viscosity of BR after thermal treatment supporting the presents results with regard of the differences in the heating process. Up to a temperature of 77 °C, physical properties of BR remained above clinically relevant minimum thresholds (Table 2). A possible explanation for the different behavior of two-component and ready-to-use formulations is the momentum of water uptake and the beginning of the setting reaction. BR is mixed with water and thermal treatment can either lead to evaporation of the water, resulting in a solid form of the sealer or to accelerated reaction kinematics enhancing the primary setting reaction of BR. Weight loss of BR was reported after thermal treatment, indicating evaporation of the contained water [11]. As TFBC and TFHF do not contain water in their premixed formulation no such effects are possible. Water uptake from surrounding structures clinically happens after the thermal process and this aspect was taken into consideration in the present study insofar as the specimens were stored in a humid environment. As both TFBC and TFHF showed no differences in setting time after the thermal treatment, the primary composition of both materials seems to be inert to thermal treatment. Thermal treatment meanwhile showed no relevant impact on the chemical composition of long-term incubated calcium silicate-based sealers. The primary composition of all sealers was inert to temperature, and water uptake in the following weeks was enough to allow a full setting reaction of the materials. TFBC and TFHF showed similar composition while the FTIR spectra of BR were different indicating a difference in the composition of premixed sealers and two-component sealers.

Water was present at all temperature levels in all specimens representing the hydration reaction of calcium silicates, which took place even after thermal treatment either from incorporated water or external sources in the humid environment. No characteristic Ca(OH)_2_ bands (O-H stretch at 3646 cm^−1^) were detected in TFHF and TFBC, while a small peak occurred in BR samples in this region [20]. After the prolonged storage, the formation of carbonates from calcium hydroxide with atmospheric CO_2_ is likely [28] as characteristic bands could be detected at 879 and from 1400 to 1500 cm^−1^ in all specimens [21,22]. Organic structures were detected in both premixed sealer formulations while not being present in BR. This finding may be due to the thickening agents or fillers used in premixed formulation to fabricate a clinically applicable sealer from a powder formulation. No such molecules must be incorporated into BR, as this sealer is mixed from powder and liquid prior to clinical application.

According to the present results and within the limitations of a laboratory simulation of thermal treatment, premixed calcium silicate-based sealer formulations can be regarded as safe for warm root canal filling techniques. The use of BR is limited to cold lateral compaction or single cone root canal filling and maybe carrier-based warm root canal filling, as these low-temperature techniques have been shown to only increase the sealer temperature to a very limited extent [17]. Still, further research is necessary to give a general recommendation about carrier-based systems.

## 5. Conclusions

Within the limitations of the study, it can be concluded that (i) thermal treatment simulating clinically relevant temperature levels and heating times does not result in relevant physical or chemical alterations to TFBC and TFHF at all temperature levels, (ii) no chemical changes can be observed with BR, and (iii) BR set immediately when heated above 77 °C for 30 s. Therefore, TFBC and TFHF can be considered suitable for warm obturation techniques, while BR should only be implemented with cold obturation techniques.

## Figures and Tables

**Figure 1 materials-14-00728-f001:**
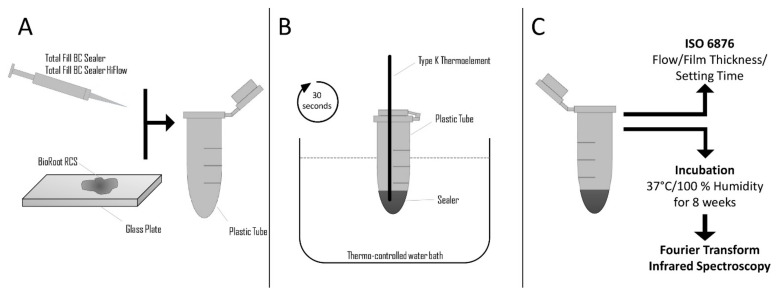
Schematic description of the experimental setup; (**A**): Total Fill BC Sealer and Total Fill BC Sealer HiFlow were dispensed directly to the vial. Mixing of BioRoot RCS and dispensing to plastic vial; (**B**): Thermal treatment of the Sealer; (**C**): Assessment of the physical and chemical characteristics of the sealers tested.

**Figure 2 materials-14-00728-f002:**
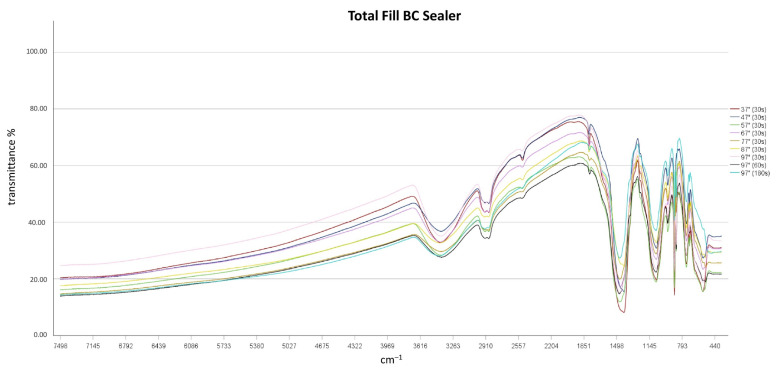
FT-IR spectroscopic plots of Total Fill BC Sealer.

**Figure 3 materials-14-00728-f003:**
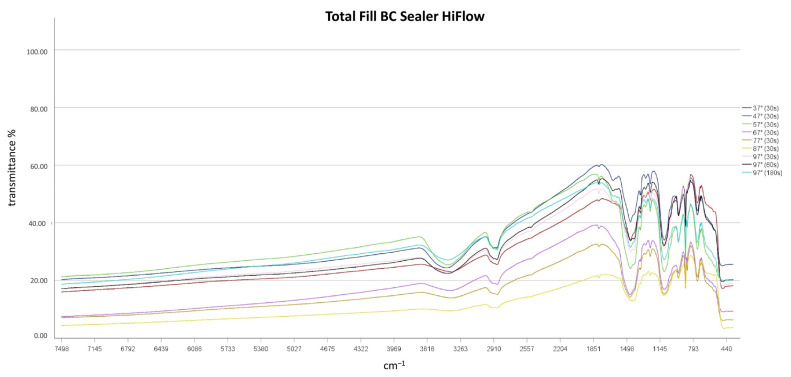
FT-IR spectroscopic plots of Total Fill BC Sealer HiFlow.

**Figure 4 materials-14-00728-f004:**
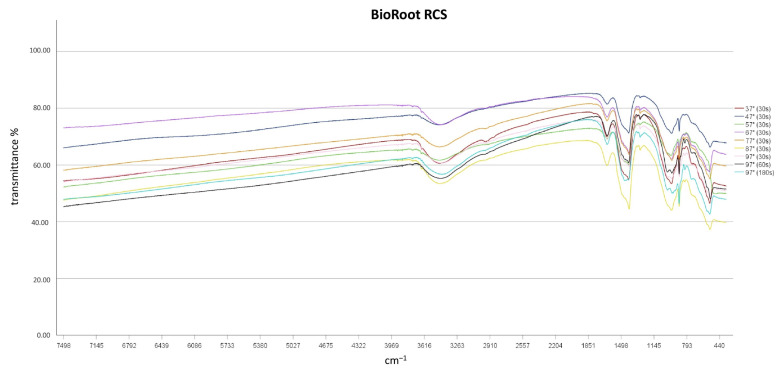
FT-IR spectroscopic plots of BioRoot RCS.

**Table 1 materials-14-00728-t001:** Composition of the sealers as provided by the manufacturers.

	Component 1	Component 2
BioRoot RCS	Powder: Tricalcium silicate, zirconium oxide, povidone	Liquid: Aqueous solution of calcium chloride and polycarboxylate
Total Fill BC Sealer	Zirconium oxide, dicalcium silicate, tricalcium silicate, calcium phosphate monobasic, calcium hydroxide, filler, thickening agents	
Total Fill BC Sealer HiFlow	Zirconium oxide, dicalcium silicate, tricalcium silicate, calcium hydroxide, filler	

**Table 2 materials-14-00728-t002:** Physical properties in accordance with ISO 6876:2012 of BioRoot RCS (means and standard deviations (SD)) after thermal treatment. Statistical analysis of setting time, film thickness and flow for BioRoot RCS was performed by Kruskal-Wallis-Test (*P* = 0.05).

	Group Number	Setting Time (min)			Film Thickness (µm)			Flow (mm)		
		Mean	SD	Different from Group Number	Mean	SD	Different from Group Number	Mean	SD	Different from Group Number
37° (30 s)	1	63.3	4.2	3	7.7	0.00	-	20.9	2.1	-
47° (30 s)	2	67.3	2.1	3,4,5	7.8	0.04	-	20.7	3.1	-
57° (30 s)	3	52.3	1.5	1,2,4,5	7.7	0.01	-	21.7	1.4	-
67° (30 s)	4	60.0	1.7	2,3	7.7	0.01	-	21.8	1.9	-
77° (30 s)	5	58.3	4.0	2,3	7.7	0.01	-	23.0	2.0	-
87° (30 s)	6	*		*	*		*	*		*
97° (30 s)	7	*		*	*		*	*		*
97° (60 s)	8	*		*	*		*	*		*
97° (180 s)	9	*		*	*		*	*		*
*P*-value				*P* = 0.025057			*P* = 0.973880			*P* = 0.609215

* Material fully set after the heat treatment process: Setting time tending towards zero, film thickness, and flow test not applicable.

**Table 3 materials-14-00728-t003:** Physical properties in accordance with ISO 6876:2012 of Total Fill BC Sealer (means and standard deviations (SD)) after thermal treatment. Statistical analysis of setting time, film thickness and flow for Total Fill BC Sealer was performed by Kruskal-Wallis-Test (*P* = 0.05).

	Group Number	Setting Time (h)			Film Thickness (µm)			Flow (mm)		
		Mean	SD	Different from Group Number	Mean	SD	Different from Group Number	Mean	SD	Different from Group Number
37° (30 s)	1	Mean	SD	2,4,5,9	7.7	0.00	4,6,4,9	24.9	0.7	-
47° (30 s)	2	19.3	1.4	1,6,7	7.7	0.00	6,7,9	25.0	0.7	-
57° (30 s)	3	17.5	0.5	4	7.7	0.00	7,9	24.6	0.3	-
67° (30 s)	4	18.1	0.1	1,3,6,7,8	7.7	0.01	1,8	24.7	0.8	-
77° (30 s)	5	17.1	0.4	1,6,7	7.7	0.00	6,7,9	25.4	1.1	-
87° (30 s)	6	17.5	0.3	2,4,5,8,9	7.7	0.01	1,2,5,8	23.9	0.5	-
97° (30 s)	7	19.2	0.4	2,4,5,9	7.7	0.01	1,2,3,5,8	28.0	2.0	-
97° (60 s)	8	19.1	0.6	4,6	7.7	0.00	4,6,7,9	24.8	1.5	-
97° (180 s)	9	18.1	0.4	1,6,7	7.7	0.00	1,2,3,5,8	26.2	2.8	-
*P*-value				*P* = 0.020231			*P* = 0.0187664			*P* = 0.213943

**Table 4 materials-14-00728-t004:** Physical properties in accordance with ISO 6876:2012 of Total Fill BC Sealer HiFlow (means and standard deviations (SD)) after thermal treatment. Statistical analysis of setting time, film thickness and flow for Total Fill BC Sealer HiFlow was performed by Kruskal-Wallis-Test (*P* = 0.05).

	Group Number	Setting Time (h)			Film Thickness (µm)			Flow (mm)		
		Mean	SD	Different from Group Number:	Mean	SD	Different from Group Number:	Mean	SD	Different from Group Number:
37° (30 s)	1	23.0	0.0	6,7,8,9	7.8	0.07	-	29.0	1.1	5,8
47° (30 s)	2	23.2	0.2	4,5,6,7,8,9	7.7	0.01	-	30.7	1.7	4,5,6,8,9
57° (30 s)	3	23.0	0.9	6,7,8,9	7.7	0.01	-	30.7	0.8	4,5,6,8,9
67° (30 s)	4	22.5	0.7	2,8,9	7.7	0.00	-	28.4	1.0	2,3,8
77° (30 s)	5	22.3	0.5	2,8	7.7	0.01	-	27.7	0.2	1,2,3,7
87° (30 s)	6	22.2	0.1	1,2,3,8	7.7	0.00	-	28.5	0.3	2,3,8
97° (30 s)	7	22.0	0.4	1,2,38	7.7	0.01	-	29.1	1.3	5,8
97° (60 s)	8	20.0	0.6	1,2,3,4,5,6,7	7.7	0.01	-	26.7	0.2	1,2,3,4,6,7
97° (180 s)	9	21.6	0.1	1,2,3,4	7.7	0.00	-	28.1	1.7	2,3
*P*-value				*P* = 0.0111865			*P* = 0.204195			*P* = 0.021105

## Data Availability

The data presented in this study are available on request from the corresponding author.

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
