# Peer review of "Physico-Chemical Investigation of Endodontic Sealers Exposed to Simulated Intracanal Heat Application: Hydraulic Calcium Silicate-Based Sealers"

_materials, 2021, doi:10.3390/ma14040728_

Round 1
Reviewer 1 Report
This study evaluated the physical properties and chemical properties of BioRoot RCS, TotalFill BC sealer, Total Fill BC sealer Hiflow after heat application.
I have a few suggestions.
1) In keywords section, it may be better to remove FT-IR.
2) Line 200-201 ‘ Also flow rates of BR were within these limitations up to a temperature level above 77℃.’
In this sentence, it may be better to describe ‘to 77℃’ or ‘at 77℃’ rather than ‘above 77℃.’
3) It would be better to write table 2 and table 5 in one table,
And to write table 3 and table 6 in one table, and to write table 4 and table 7 in one table, respectively.
4)line 230-231
It may be better to revise ‘FT-IR spectroscopy did not reveal any changes in the spectroscopic plots indicating that no chemical changes of the sealers occurred’
To ‘FT-IR spectroscopy did not reveal any changes in the spectroscopic plots indicating that no chemical changes of the three sealers occurred’
5) line 271
It may be better to remove ‘Fourier transform infrared spectroscopy’ , because the abbreviation was already used in the manuscript.
Author Response
Dear Reviewer,
thank you for your comments.
In the following you find a point-to-point answer:
1) Keyword removed according to the comment
2) wording revised according to the comment
3) The tables 2 and 5, 3 and 6, 4 and 7 were merged as demanded in the comment
4) wording revised according to the comment
5) wording revised according to the comment
Reviewer 2 Report
Dear Authors,
The article presented is great of interest.
However, it presents some criticality:
Abstract: Abstract should be revised and re written to make it more appealing. Please follow the instruction for authors. “The abstract should be a total of about 200 words maximum. The abstract should be a single paragraph and should follow the style of structured abstracts, but without headings: 1) Background: Place the question addressed in a broad context and highlight the purpose of the study; 2) Methods: Describe briefly the main methods or treatments applied. Include any relevant preregistration numbers, and species and strains of any animals used. 3) Results: Summarize the article's main findings; and 4) Conclusion: Indicate the main conclusions or interpretations. The abstract should be an objective representation of the article: it must not contain results which are not presented and substantiated in the main text and should not exaggerate the main conclusions.” Your abstract presents more than 300 words. Please revise
Keywords: To ensure a properly research in medical databases, use MeSH terms to find the keywords.
Introduction:
“Higher sealer proportions were no longer regarded disadvantageous as biological and antibacterial effects of calcium-silicate based sealers are thought to enhance the success of root canal treatment. Due to a low compression of the root canal filling material and sealer during single cone root canal filling, this technique is regarded incapable to fill complex root canal systems. In simpler root canal anatomies, it might allow sufficient root canal filling.” These sentences should be revised. Moreover please cite some studies that could sustain these sentences.
To ensure a proper presentation of the topic of the article I would like to suggest to insert this article in the introduction section. It could be helpful to complete the comprehension of these new cements. “Donfrancesco, O., Seracchiani, M., Morese, A., Ferri, V., Nottola, S.A., Relucenti, M., Gambarini, G. et al. 2020, "Analysis of Stability in Time of Marginal Adaptation of Endosequence Root Repair Material on Biological Samples", Dental Hypotheses, vol. 11, no. 1, pp. 11-15.”
The aim of the study should be well presented and stated. Please reformulate it and re control the premises.
Please revise entirely this section, revising not only English spell but also the fluency.
Materials And Methods:
Methodology is well presented and written.
However, the statistical analysis section could be implemented:
You should indicate what software do you use to collect your data (excel, SPSS or others).
Could you please indicate why have you chosen the Kruskal-Wallis-Test? How do you demonstrate that your specimens are not normally distributed. What is the kurtosis value? Have you performed descriptive statistic to your specimens?
Results:
I would like to propose to reduce the numbers of the tables. If is it possibile please provide a more schematic one.
Discussion:
“To investigate the long-term effect of thermal treatment on calcium silicate based sealers, specimens were stored for 8 weeks. Calcium silicate-based materials have a prolonged setting reaction [13] and, especially in the case of premixed sealer formulations, communicate with the environment to incorporate surrounding water during the setting reaction. Therefore, further to the short-term effect of heat on possible destruction of the chemical composition the long-term effect of heat on the setting profile is of scientific interest.” You need to be more supported by current literature. Please cite other studies that could support your findings.
“The primary composition of all sealers was inert to temperature and water uptake in the following weeks was enough to allow a full setting reaction of the materials. TFBC and TFHF showed similar composition while the FTIR spectra of BR were different indicating a difference in the composition of premixed sealers and two component sealers.” As described above, you should insert more studies that could support your findings.
At the end of the discussion section you should indicate the limitations of the study.
Conclusion:
Conclusion section should be completely revised and make it more appealing.
English spell revision is necessary.
The layout of the entire manuscript should be corrected following the instruction for authors.
The font used for the references should be the same of the entire manuscript.
Best Regards.

Author Response
Dear Reviewer,
thank you for your comments.
In the following you find a point-to-point answer:
1) Abstract: The Abstract was shortened to 200 words.
2) KeyWords: Matching MeSh Terms were added to the keywords
3) Intro: Literature was added to support the statement.
4) Intro: The suggested citation was added to the manuscript.
5) Intro: Aim and hypothesis are given in the last Paragraph of the introduction.
6) M&M: Information About data Collection was added.
7) Due to the sample size indicated by ISO 6876, test on normal Distribution is not possible. Kruskal-Wallis Test is suitable for small sample sizes.
8) Results: The number of tables was reduced as suggested by Reviewer 1
9) Discussion: Literature was added to support the statement.
10) Discussion: This is a major finding of the study presented in the results section and new information. Therefore a citation of other literature is not possible.
11) Discussion: Information about the limitations were added.
12) Conclusions: the section was revised to make it more appealing
Language, Format and references were revised.
Round 2
Reviewer 2 Report
Dear Authors,
please re send the manuscript with the correct references.
In the present version I can find only 22 references instead of the 28 cited in the manuscript.
Moreover please provide a correct layout of the manuscript.
Furthermore I would like to suggest to resend the author's reply with a .doc file point by point.
Please revise the materials and methods and results sections making it more fluent and appealing
Best Regards
Author Response
Dear reviewer,
thank you for your comments. In the following you find a point-to-point answer:
please re send the manuscript with the correct references.
In the present version I can find only 22 references instead of the 28 cited in the manuscript.
Something went wrong the Microsoft Word correction function and the pdf building. The correct reference section was added to the manuscript.
Moreover please provide a correct layout of the manuscript.#
The layout is according to the submission guidelines. Also, the journal allows free format submission.
Furthermore I would like to suggest to resend the author's reply with a .doc file point by point.
A .doc file was uploaded.
Please revise the materials and methods and results sections making it more fluent and appealing
In our opinion these sections are presented well. No such issues were raised in the first round or by the other reviewer.
